# Short-term molecular and cellular effects of ischemia/reperfusion on vascularized lymph node flaps in rats

Florian S. Frueh[1,2]*, Bijan Jelvani[1], Claudia Scheuer[1], Christina Körbel[1], Bong-Sung Kim[2], Pietro Giovanoli[2], Nicole Lindenblatt[2], Yves Harder[3,4], Emmanuel Ampofo[1], Michael D. Menger[1], Matthias W. Laschke[1]

1 Institute for Clinical and Experimental Surgery, Saarland University, Homburg, Saar, Germany, 2 Department of Plastic Surgery and Hand Surgery, University Hospital Zurich, University of Zurich, Zurich, Switzerland, 3 Department of Plastic, Reconstructive and Aesthetic Surgery, Ente Ospedaliero Cantonale (EOC), Lugano, Switzerland, 4 Faculty of Biomedical Sciences, Università della Svizzera Italiana (USI), Lugano, Switzerland

* florian.frueh@usz.ch

**Data Availability Statement:** All relevant data are within the manuscript and its Supporting Information files.

## Abstract

Vascularized lymph node (VLN) transfer is an emerging strategy to re-establish lymphatic drainage in chronic lymphedema. However, the biological processes underlying lymph node integration remain elusive. This study introduces an experimental approach facilitating the analysis of short-term molecular and cellular effects of ischemia/reperfusion on VLN flaps. Lymph node flaps were dissected pedicled on the lateral thoracic vessels in 44 Lewis rats. VLN flaps were exposed to 45 or 120 minutes ischemia by *in situ* clamping of the vascular pedicle with subsequent reperfusion for 24 hours. Flaps not exposed to ischemia/reperfusion served as controls. Lymph nodes and the perinodal adipose tissue were separately analyzed by Western blot for the expression of lymphangiogenic and angiogenic growth factors. Moreover, morphology, microvessel density, proliferation, apoptosis and immune cell infiltration of VLN flaps were further assessed by histology and immunohistochemistry. Ischemia for 120 minutes was associated with a markedly reduced cellularity of lymph nodes but not of the perinodal adipose tissue. In line with this, ischemic lymph nodes exhibited a significantly lower microvessel density and an increased expression of VEGF-D and VEGF-A. However, VEGF-C expression was not upregulated. In contrast, analyses of the perinodal adipose tissue revealed a more subtle decrease of microvessel density, while only the expression of VEGF-D was increased. Moreover, after 120 minutes ischemia, lymph nodes but not the perinodal adipose tissue exhibited significantly higher numbers of proliferating and apoptotic cells as well as infiltrated macrophages and neutrophilic granulocytes compared with non-ischemic flaps. Taken together, lymph nodes of VLN flaps are highly susceptible to ischemia/reperfusion injury. In contrast, the perinodal adipose tissue is less prone to ischemia/reperfusion injury.

**Funding:** YES 1. FSF was supported by Deutsche Gesellschaft für Lymphologie (DGL) 2. Grant number: Fo0315 3. https://www.dglymph.de/aktuelles 4. No, sponsor not involved in research.

**Competing interests:** The authors have declared that no competing interests exist.

## Introduction

The lymphatic system is essential for tissue fluid homeostasis in higher vertebrates [1]. Furthermore, it is involved in the regulation of immunosurveillance and the absorption of dietary fats [2]. The lymphatic vasculature consists of capillaries, precollecting vessels and collecting lymphatic trunks, which form a complex network with interposed lymph nodes. Its disruption may result in lymphedema, a condition characterized by limb swelling, chronic interstitial inflammation and connective or adipose tissue deposition [3,4]. In developed countries, cancer-related lymphedema represents the most common form. It is estimated that ~ 1 out of 6 patients treated for a solid tumor develops lymphedema related to type and extent of treatment, anatomical location, and length of follow-up [5]. Particularly breast cancer is associated with high rates of secondary lymphedema. In fact, > 20% of cancer survivors undergoing axillary lymph node dissection develop arm lymphedema [6]. Given the livelong course of the disease, lymphedema is a major socio-economic burden [7].

Complex decongestive therapy is still considered the gold standard in the management of lymphedema [8]. This treatment may avoid disease progression, but it cannot offer a cure. In the last decades, reconstructive microsurgical procedures, such as the transfer of vascularized lymph node (VLN) flaps [9], the connection of afferent lymphatic vessels to the venous system (lymphovenous anastomosis) [10], and the transplantation of lymphatic vessels [11] have been introduced. These techniques aim at a functional reconstruction of the lymphatic pathway and, thus, have a curative potential. Recent evidence indicates that the transfer of VLN flaps may be the most powerful approach to re-establish a functional lymphatic drainage [12,13]. These flaps contain a variable number of lymph nodes and perinodal adipose tissue. Since the introduction of VLN flaps in 1979 [14], numerous studies have analyzed the function of transplanted lymph nodes and revealed that they predominantly act as a "lymphatic pump", draining the interstitial lymph through intranodal lymphovenous connections to the venous circulation [13,15]. According to this theory, the efficacy of VLN flaps is positively correlated with their content of lymph nodes [16]. However, this drainage can only take place if the transplanted lymph nodes are functionally integrated into the surrounding host tissue. Importantly, the biological processes underlying functional lymph node integration are complex and not well understood.

Previous preclinical and clinical studies reporting mechanisms of VLN flap integration predominantly focused on the transplanted lymph nodes [13,17]. However, it may be assumed that not only the transplanted lymph nodes but also the perinodal adipose tissue contributes to the restoration of lymphatic function after VLN flap surgery. In the present study, we introduce an experimental model for the assessment of short-term molecular and cellular effects of intraoperative ischemia/reperfusion (IR) on VLN flaps. For this purpose, axillary VLN flaps were dissected in a rat model and exposed to IR. The animal model was validated by means of ultrasound and photoacoustic imaging. Histological, immunohistochemical and Western blot analyses were performed to quantify growth factor expression, microvessel density, cell proliferation, immune cell infiltration, oxidative stress and apoptotic cell death in lymph nodes and the perinodal adipose tissue of VLN flaps.

## Results

### Animal model

VLN flaps were dissected pedicled on the lateral thoracic vessels in the right axilla of Lewis rats (Fig 1A–1C). After complete dissection, the vascular pedicle of the flaps was clamped for 45 minutes (IR-45) or 120 minutes (IR-120) using microvascular clamps. After releasing the

**Fig 1. The axillary VLN flap model. A** T-shaped skin incision in the right axilla. **B** The pectoralis major muscle is retracted cranio-medially (black arrow) and the lateral thoracic artery (red arrowhead) and vein (blue arrowhead) are identified. Dotted line and asterisk = VLN flap. **C** Completely dissected VLN flap containing lymph nodes (broken lines) and perinodal adipose tissue (asterisk). Vascular pedicle with lateral thoracic artery (red arrowhead), axillary artery (double red arrowhead), lateral thoracic vein (blue arrowhead), and axillary vein (double blue arrowhead). **D** Separation of lymph nodes and perinodal adipose tissue for Western blot analyses. Scale bars: **A** and **B** = 6 mm, **C** = 3 mm. VLN = vascularized lymph node.

clamps, the skin incisions were closed and the animals were observed for 24 hours. There were no complications with wound healing and all animals showed unrestricted mobilization of the operated limb. Following the IR protocol, VLN flaps were sampled for immunohistochemistry or Western blot (Fig 1D). For the latter, the lymph nodes of individual flaps were microsurgically excised from the perinodal tissue using a stereomicroscope.

## Validation of animal model

For an objective *in vivo* assessment of IR, VLN flaps of the IR-45 and IR-120 groups were analyzed by means of ultrasound and photoacoustic imaging (Fig 2A–2P). During ischemia, $sO_2$ significantly decreased in the lymph nodes of both groups by ~ 40% when compared to baseline measurements (Fig 2A–2F, 2N and 2P). After reperfusion, $sO_2$ rapidly returned to baseline levels (Fig 2G–2L, 2N and 2P). There was no statistically significant difference between $sO_2$ levels of IR-45 and IR-120 lymph nodes (Fig 2N and 2P). In addition, $HbT/mm^3$ was constant throughout the course of the experiments (Fig 2M and 2O). Moreover, the lymph nodes of both groups exhibited comparable $HbT/mm^3$ levels after 24 hours of reperfusion (Fig 2M and 2O). This indicates that the observed changes in tissue oxygenation were not caused by different tissue hemoglobin levels but reduced oxygenated hemoglobin. Finally, stereomicroscopic imaging revealed a dilated perinodal vascular network with small capsular hemorrhages in both ischemia groups (Fig 2Q–2T).

## Morphology of VLN flaps

Furthermore, the impact of ischemia on the morphology of axillary VLN flaps was assessed by means of histological and immunohistochemical staining (Fig 3). All groups exhibited a comparable lymph node structure with subcapsular, intermediate and medullary sinuses as well as follicles (Fig 3A–3G), characterized by a LYVE-1+ lymphatic network separating these areas (Fig 3B, 3C, 3E and 3G). However, ischemia for 120 minutes was associated with a significant decrease of lymph node cellularity (Fig 3H–3K). In contrast, the perinodal adipose tissue of the IR-45 and IR-120 groups did not exhibit lower cell counts when compared to the control group (Fig 3L).

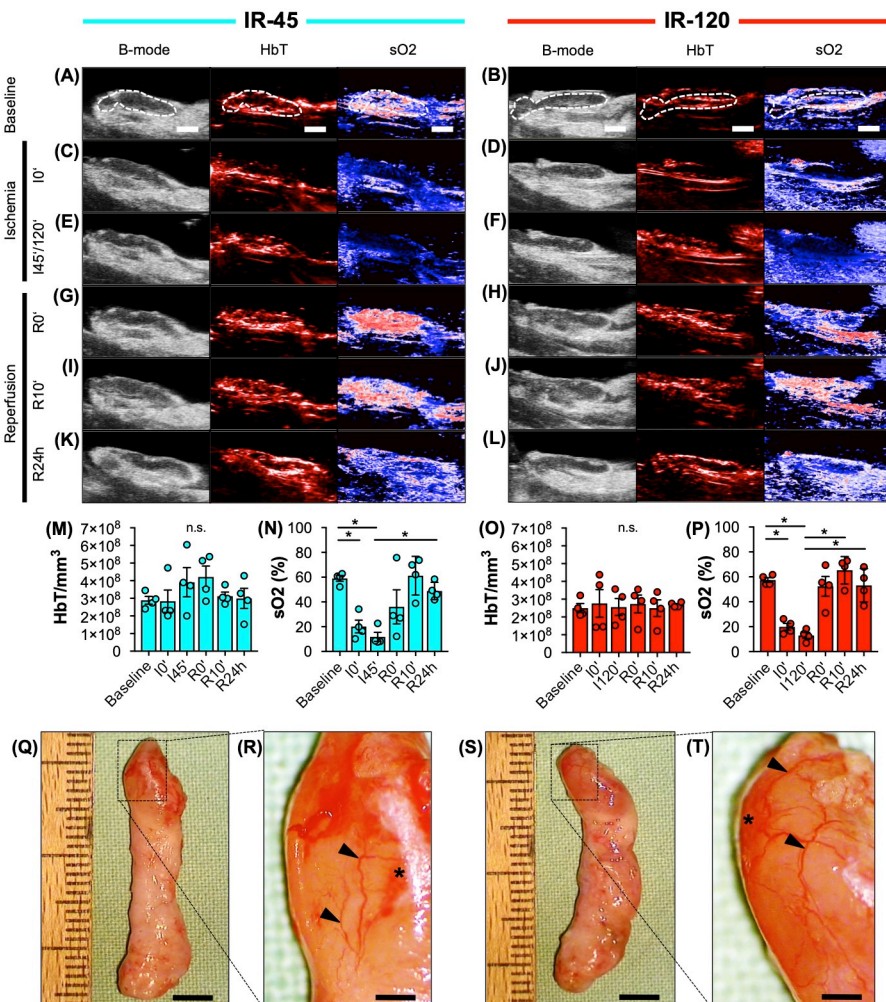

**Fig 2. Animal model validation. A-L** Ultrasound (B-mode) and photoacoustic (HbT, sO₂) imaging of IR-45 and IR-120 axillary VLN flaps before (**A** and **B**), during (**C-F**) and after (**G-L**) ischemia (broken line = lymph node). **M-P** Total hemoglobin (HbT/mm³) and oxygen saturation (sO₂ (%)) of IR-45 (**M** and **N**, cyan) and IR-120 (**O** and **P**, red) lymph nodes at baseline, beginning of ischemia (I0'), after 45 minutes (I45') or 120 minutes (I120') of ischemia as well as directly (R0'), 10 minutes (R10') and 24 hours (R24h) after reperfusion. Mean ± SEM, n = 4, *P < 0.05, n.s. = not significant. **Q-T** Stereomicroscopic images of IR-45 (**Q** and **R**) and IR-120 (**S** and **T**) VLN flaps. Higher magnification (**R** and **T** = inserts of **Q** and **S**) reveals a dilated vascular network (arrowheads) with capsular hemorrhages (asterisks). Scale bars: **A-L** = 2 mm, **Q** and **S** = 4.5 mm, **R** and **T** = 900 μm. IR = ischemia/reperfusion, VLN = vascularized lymph node.

## Microvascular network analysis

VLN flaps were further investigated for alterations of their microvascular network after IR injury. For this purpose, sections were stained with the endothelial cell marker CD31 and the microvessel density of lymph nodes as well as the perinodal adipose tissue was quantified (Fig 4). Lymph nodes of the IR-45 and IR-120 groups exhibited a significantly reduced density of CD31⁺ microvessels when compared to the control group (Fig 4E). In the perinodal adipose tissue, the overall number of microvessels was lower when compared to that within the lymph nodes (Fig 4F). Moreover, the negative IR effect on microvessel density was less pronounced when compared to the changes observed in the nodal component of the flaps (Fig 4F).

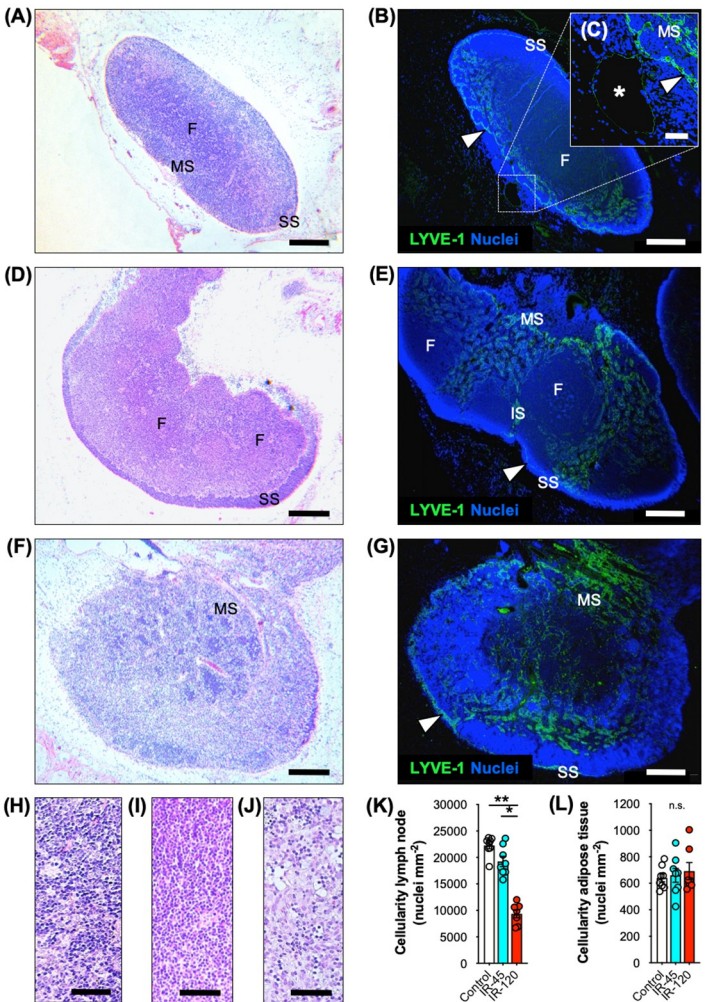

**Fig 3. Structural alterations of VLN flaps after IR injury. A-G** HE-stained (**A, D** and **F**) and immunohistochemical (**B, C, E** and **G**) sections of control (**A-C**), IR-45 (**D** and **E**) and IR-120 (**F** and **G**) VLN flaps. SS = subcapsular sinus, IS = intermediate sinus, MS = medullary sinus, F = follicle. Arrowhead = LYVE-1$^+$ cell-lined sinus. **C** (insert of **B**) Asterisk = main efferent lymphatic vessel. **H-J** HE-stained section of follicles of control (**H**), IR-45 (**I**) and IR-120 (**J**) lymph nodes. **K** Lymph node cellularity (nuclei mm$^{-2}$). Mean ± SEM, n = 7–8, $^*P < 0.05$, $^{**}P < 0.001$. **L** Adipose tissue cellularity (nuclei mm$^{-2}$). Mean ± SEM, n = 7–8, n.s. = not significant. Scale bars: **A, B, D-G** = 300 μm, **C** = 40 μm, **H-J** = 60 μm. HE = hematoxylin and eosin, LYVE = lymphatic vessel endothelial hyaluronan receptor, VLN = vascularized lymph node.

## Expression of lymphangiogenic and angiogenic growth factors

To examine the effects of IR on lymphangiogenesis and angiogenesis, the marker proteins vascular endothelial growth factor (VEGF)-C, VEGF-D and VEGF-A were analyzed by means of Western blot (Fig 5A–5D). For protein expression analyses, lymph nodes and perinodal adipose tissue of individual flaps were sampled separately. We found a significantly increased expression of the lymphangiogenic growth factor VEGF-D in lymph nodes (Fig 5C) as well as perinodal adipose tissue (Fig 5D) after 120 minutes ischemia when compared to samples of the control and IR-45 group. Surprisingly, VEGF-C expression was not upregulated after 120 minutes ischemia. Furthermore, the expression of the angiogenic growth factor VEGF-A was significantly higher in the lymph nodes of the IR-120 group but not in the corresponding

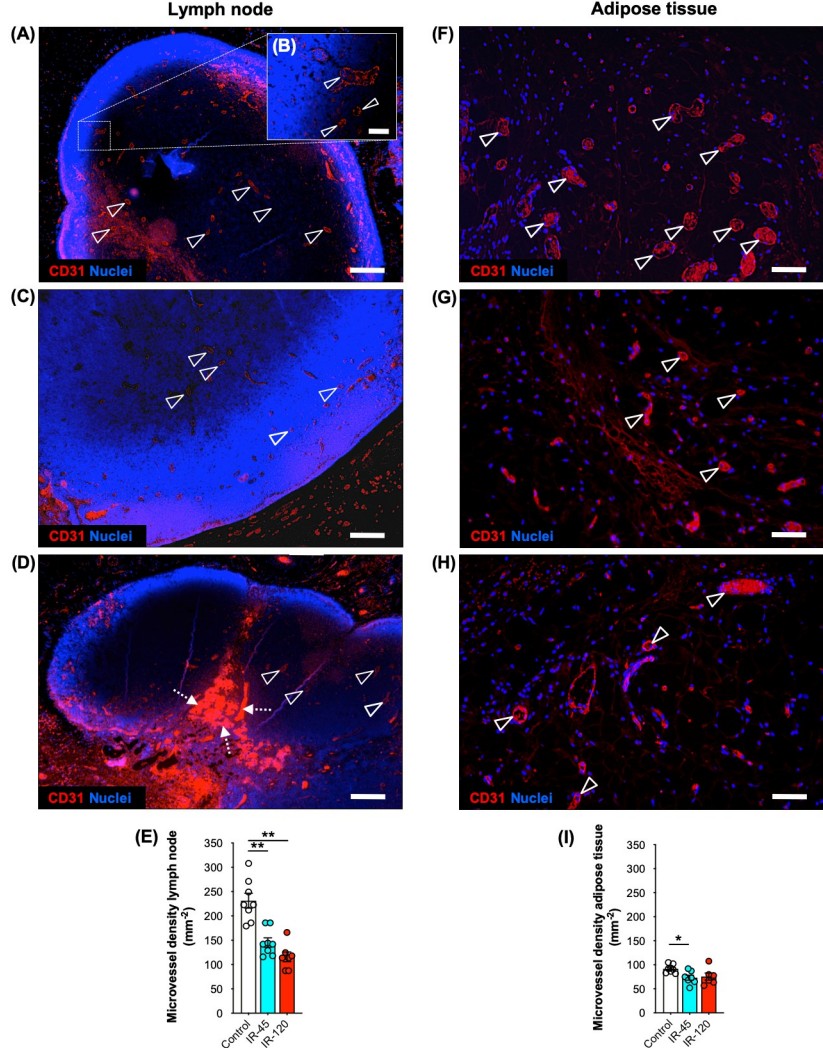

**Fig 4. Microvessel density of VLN flaps after IR injury. A-D** Immunohistochemical sections of control (**A** and **B**), IR-45 (**C**) and IR-120 (**D**) VLN flaps illustrating CD31$^+$ microvessels (arrowheads) in lymph nodes. **B** (insert of **A**) CD31$^+$ paracortical microvessels (arrowheads). Dashed arrows in **D** = local hemorrhages. **E** Lymph node microvessel density (mm$^{-2}$). Mean ± SEM, n = 7–8, $^{**}P < 0.001$. **F-H** Immunohistochemical sections of control (**F**), IR-45 (**G**) and IR-120 (**H**) VLN flaps illustrating CD31$^+$ microvessels (arrowheads) in the perinodal adipose tissue. **I** Adipose tissue microvessel density (mm$^{-2}$). Mean ± SEM, n = 7–8, $^*P < 0.05$. Scale bars: **A, C** and **D** = 250 μm, **B** = 40 μm, **F-H** = 60 μm. VLN = vascularized lymph node.

perinodal adipose tissue (Fig 5C and 5D). The original Western blots are provided in the supplementary information.

## Expression of iNOS, eNOS and p-eNOS

Additional Western blot analyses were performed to analyze the expression of the inducible nitric oxide (NO)-synthase (iNOS) and the constitutively expressed endothelial NO-synthase (eNOS) and its phosphorylated form phospho-endothelial NO-synthase (p-eNOS). We found that neither iNOS nor p-eNOS/eNOS expression were significantly upregulated in ischemic lymph nodes when compared to lymph nodes of the control group (Fig 5E, 5G and 5H). In contrast, perinodal adipose tissue undergoing 120 minutes ischemia exhibited a significantly

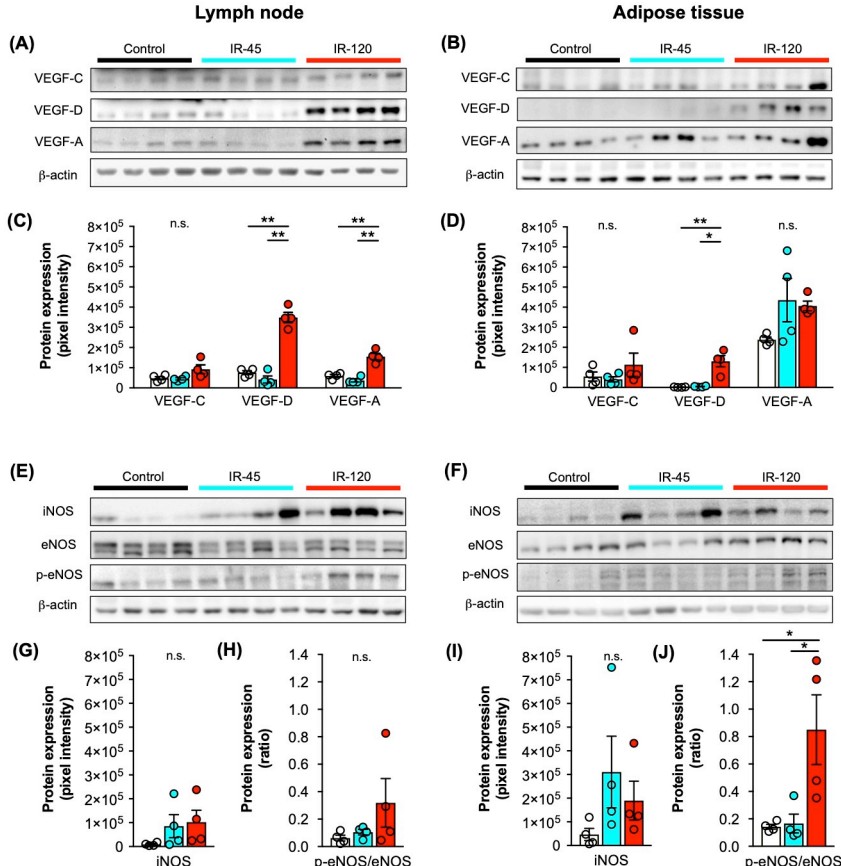

**Fig 5. Growth factor and NOS expression of VLN flaps after IR injury. A-D** Western blots (**A** and **B**) with quantification (**C** and **D**) of VEGF-C, VEGF-D and VEGF-A expression of control (white), IR-45 (cyan) and IR-120 (red) VLN flaps. Bands were cropped from different parts of the same gel. Panels **A** and **C** represent analyses of lymph nodes and panels **B** and **D** analyses of adipose tissue. Mean ± SEM, n = 4, $^*P < 0.05$, $^{**}P < 0.001$, n.s. = not significant. **E-J** Western blots (**E** and **F**) with quantification of iNOS (**G** and **I**) and p-eNOS/eNOS (**H** and **J**) expression of control (white), IR-45 (cyan) and IR-120 (red) VLN flaps. Bands were cropped from different parts of the same gel. Panels **E, G** and **H** represent analyses of lymph nodes and panels **F, I** and **J** analyses of adipose tissue. Mean ± SEM, n = 4, $^*P < 0.05$, n.s. = not significant. IR = ischemia/reperfusion, iNOS = inducible nitric oxide synthase (NOS), eNOS = endothelial NOS, p-eNOS = phospho-eNOS, VEGF = vascular endothelial growth factor, VLN = vascularized lymph node.

increased p-eNOS/eNOS ratio when compared to adipose tissue of the control and IR-45 group, indicating adipose tissue susceptibility to IR injury (Fig 5F, 5I and 5J). The original Western blots are provided in the supplementary information.

## Oxidative stress, cell proliferation, cell apoptosis

To quantify oxidative stress, cellular proliferation and apoptotic cells in ischemic VLN flaps sections were stained with antibodies against the heat-shock protein heme oxygenase-1 (HO-1), proliferating cell nuclear antigen (PCNA) as well as cleaved caspase-3 (Casp-3). Immuno-histochemical analyses revealed a significantly higher fraction of cells expressing HO-1 in lymph nodes of the IR-120 group when compared to control and IR-45 lymph nodes (Fig 6A–6D). Moreover, lymph nodes contained a markedly higher number of PCNA$^+$ cells (Fig 6E–6H) and Casp-3$^+$ cells (Fig 6I–6L) after 120 minutes of ischemia. The majority of the proliferating and apoptotic cells were lymphocytes, as verified by microscopic assessment of cell

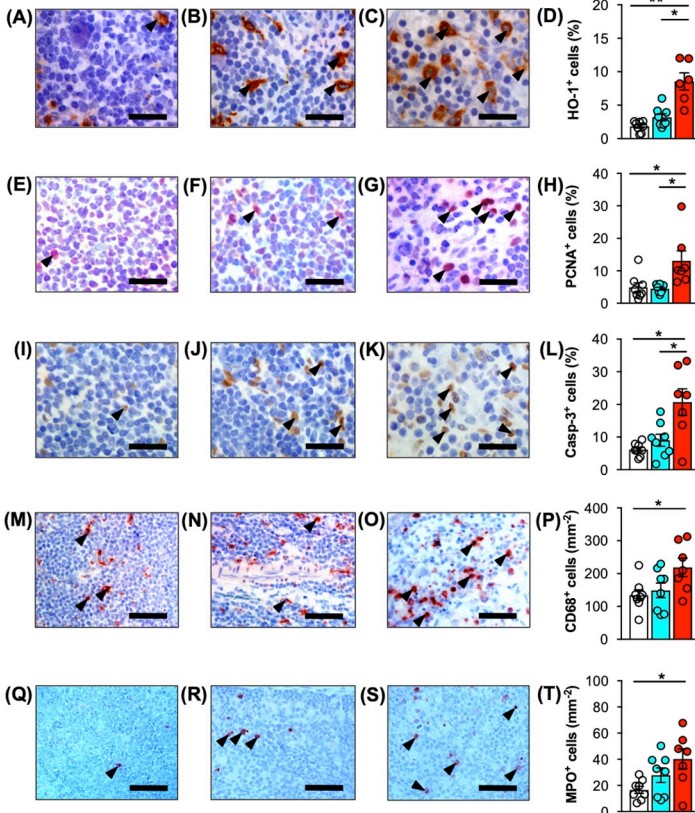

**Fig 6. Immunohistochemical analyses of lymph nodes after IR injury. A-D** Immunohistochemical detection of HO-1+ cells (arrowheads) in control (**A**), IR-45 (**B**) and IR-120 (**C**) lymph nodes. **D** Quantitative analysis of HO-1+ cells (%). **E-H** Immunohistochemical detection of PCNA+ cells (arrowheads) in control (**E**), IR-45 (**F**) and IR-120 (**G**) lymph nodes. **H** Quantitative analysis of PCNA+ cells (%). **I-L** Immunohistochemical detection of Casp-3+ cells (arrowheads) in control (**I**), IR-45 (**J**) and IR-120 (**K**) lymph nodes. **L** Quantitative analysis of Casp-3+ cells (%). **M-P** Immunohistochemical detection of CD68+ macrophages (arrowheads) in control (**M**), IR-45 (**N**) and IR-120 (**O**) lymph nodes. **P** Quantitative analysis of CD68+ cells (mm-2). **Q-T** Immunohistochemical detection of MPO+ neutrophilic granulocytes (arrowheads) in control (**Q**), IR-45 (**R**) and IR-120 (**S**) lymph nodes. **T** Quantitative analysis of MPO+ cells (mm-2). Mean ± SEM, n = 6–8, *$P < 0.05$, **$P < 0.001$. Scale bars: **A-C, E-G** and **I-K** = 25 μm, **M-O** and **Q-S** = 50 μm. Casp-3 = cleaved caspase-3, HO = heme oxygenase, IR = ischemia/reperfusion, MPO = myeloperoxidase, PCNA = proliferating cell nuclear antigen, VLN = vascularized lymph node.

morphology. Surprisingly, immunohistochemical investigations of the perinodal adipose tissue did not reveal significantly increased oxidative stress, cell proliferation as well as apoptotic cell death after 45 and 120 minutes of ischemia, as indicated by HO-1, PCNA and Casp-3 stainings (Fig 7A–7L).

## Immune cell infiltration

Finally, for the quantification of immune cell infiltration of VLN flaps sections were stained with CD68 and myeloperoxidase (MPO) for the detection of macrophages and neutrophilic granulocytes, respectively. Prolonged ischemia of VLN flaps was associated with an increased immune cell infiltration in the nodal flap component. Accordingly, lymph nodes of the IR-120 group exhibited a significantly higher number of CD68+ macrophages (Fig 6M–6P) and MPO+ neutrophilic granulocytes (Fig 6Q–6T) when compared to control lymph nodes. In contrast, analyses of the perinodal adipose tissue did not show significantly higher numbers of

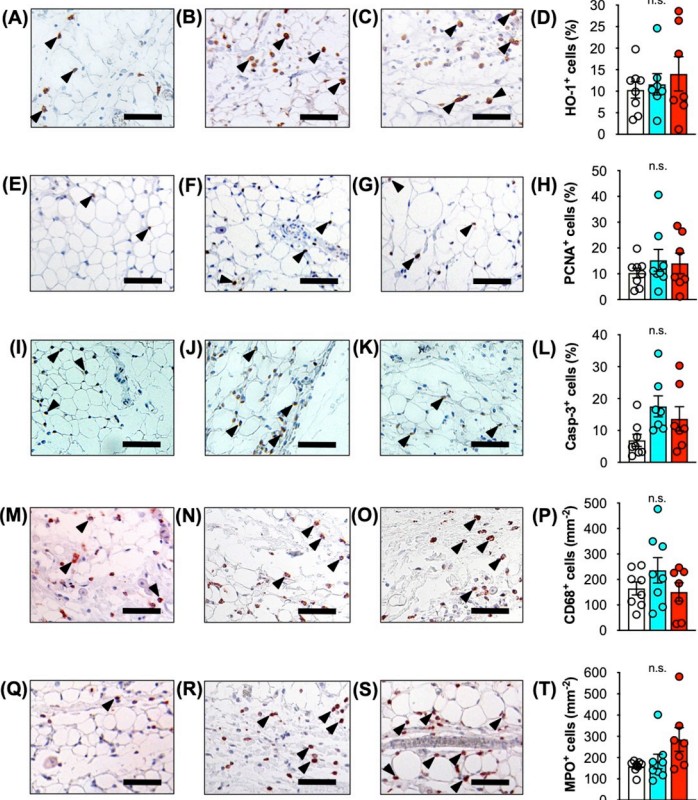

**Fig 7. Immunohistochemical analyses of adipose tissue after IR injury. A-D** Immunohistochemical detection of HO-1$^+$ cells (arrowheads) in perinodal adipose tissue of control (**A**), IR-45 (**B**) and IR-120 (**C**) VLN flaps. **D** Quantitative analysis of HO-1$^+$ cells (%). **E-H** Immunohistochemical detection of PCNA$^+$ cells (arrowheads) in perinodal adipose tissue of control (**E**), IR-45 (**F**) and IR-120 (**G**) VLN flaps. **H** Quantitative analysis of PCNA$^+$ cells (%). **I-L** Immunohistochemical detection of Casp-3$^+$ cells (arrowheads) in perinodal adipose tissue of control (**I**), IR-45 (**J**) and IR-120 (**K**) VLN flaps. **L** Quantitative analysis of Casp-3$^+$ cells (%). **M-P** Immunohistochemical detection of CD68$^+$ macrophages (arrowheads) in perinodal adipose tissue of control (**M**), IR-45 (**N**) and IR-120 (**O**) VLN flaps. **P** Quantitative analysis of CD68$^+$ cells (mm$^{-2}$). **Q-T** Immunohistochemical detection of MPO$^+$ neutrophilic granulocytes (arrowheads) in perinodal adipose tissue of control (**Q**), IR-45 (**R**) and IR-120 (**S**) VLN flaps. **T** Quantitative analysis of MPO$^+$ cells (mm$^{-2}$). Mean ± SEM, n = 7–8, n.s. = not significant. Scale bars = 50 µm. Casp-3 = cleaved caspase-3, HO = heme oxygenase, IR = ischemia/reperfusion, MPO = myeloperoxidase, PCNA = proliferating cell nuclear antigen, VLN = vascularized lymph node.

infiltrating macrophages and neutrophilic granulocytes in the IR-45 and IR-120 groups when compared to the non-ischemic control (Fig 7M–7T).

## Discussion

Cancer-related chronic lymphedema is a highly prevalent disease with an underestimated socio-economic burden. In the last decades, promising microsurgical treatment strategies have been introduced with the aim to restore physiological lymphatic drainage. Among these, VLN flaps exhibit a particularly high potential to tackle the pathophysiological changes of chronic lymphedema. Despite recent hallmark studies describing the mechanism how VLN flaps ameliorate lymphatic drainage [13,15], the biological processes underlying flap integration remain a subject of debate.

In the present investigation, we analyzed the short-term effect of IR injury on VLN flaps. To prevent challenging repetitive monitoring of rodent anastomoses, we used an IR model

with temporary clamping of the vascular pedicle. In clinical practice, intraoperative ischemia of flaps commonly ranges between 45 and 120 minutes. Consequently, we applied ischemia for 45 and 120 minutes to generate data of translational interest. Photoacoustic imaging revealed a consistent reduction of oxygen saturation in VLN flaps after 45 and 120 minutes ischemia, indicating a high reliability of the model. We did not find a compensatory hyper-oxygenation at the onset of reperfusion. Furthermore, stereomicroscopic assessment of VLN flaps after 45 and 120 minutes ischemia did not reveal different alterations of the superficial vascular network. However, detailed histological and immunohistochemical analyses showed that 120 minutes ischemia may already be critical for the physiological lymph node architecture [18]. In fact, we found a significantly lower lymph node cellularity after 120 minutes ischemia when compared to the control and IR-45 samples, leading to a ~ 50% reduction of cell nuclei in lymph nodes undergoing prolonged ischemia. In contrast, the perinodal adipose tissue of VLN flaps undergoing IR injury was not characterized by lower cell counts when compared to control samples, indicating a higher susceptibility of the nodal component to ischemic damage.

Functional lymph node integration of VLN flaps critically depends on the expression of lymphangiogenic growth factors, such as VEGF-C and VEGF-D [19]. For instance, exogenous VEGF-C enhances lymphatic vessel formation and function and is crucial to preserve lymph node histology of VLN in pigs [20]. Transferred lymph nodes have been identified as an endogenous source for lymphangiogenic growth factor expression [17]. Surprisingly, little effort has been directed towards the clarification of the function of the adipose tissue within VLN flaps. The perinodal adipose tissue contains a dense lymphatic network and is rich of stem cells and inflammatory cells, which may support lymph node integration after VLN flap transfer. According to this assumption, Aschen et al. reported high VEGF-C expression in adipose tissue of the recipient site after avascular lymph node transplantation in mice [21]. Based on these findings, we speculated that the perinodal adipose tissue of VLN flaps may be equally or even more important for intraoperative endogenous growth factor expression compared with the nodal component.

To test this hypothesis, Western blot analyses were performed to evaluate the immediate effect of IR on growth factor expression in VLN flaps. We found a markedly increased expression of VEGF-D in lymph nodes and perinodal adipose tissue of VLN flaps after 120 minutes ischemia, whereas VEGF-C expression was not upregulated in this group. Furthermore, we observed an upregulated VEGF-A expression in lymph nodes after 120 minutes ischemia and a high VEGF-A expression in the perinodal adipose tissue of all groups. These findings are important because VEGF-D and VEGF-A overexpression can lead to blood and lymphatic vessel hyperpermeability with seroma formation through VEGF receptor-2 (VEGFR-2) signaling [22]. In line with our findings, previous clinical studies also revealed increased VEGF-D levels in axillary seroma fluid after VLN flap surgery [23,24]. However, these studies found even higher levels of VEGF-C, which is commonly accepted as the most important pro-lymphangiogenic growth factor concerning reconstructive lymphatic surgery [22]. Importantly, analyzing postoperative seroma fluid does not allow identifying the source of growth factor expression. Hence, even though limited by a small sample size, our analysis contributes to the understanding of perioperative lymphangiogenic growth factor expression during VLN flap surgery.

Noteworthy, the increased VEGF-A expression in IR-exposed lymph nodes was not associated with a higher microvessel density, as indicated by additional immunohistochemical analyses. In fact, lymph nodes undergoing 45 and 120 minutes of ischemia even contained less CD31$^+$ microvessels when compared to non-ischemic controls. From a biological point of view, this finding is not surprising because IR-related tissue damage leads to the loss of microvascular structures and local hypoxia. Consequently, angiogenic growth factors such as

VEGF-A are upregulated, promoting angiogenic neovessel growth. However, angiogenesis is a time-consuming process with a slow growth rate of new microvessels of ~ 5 μm per hour [25]. Hence, 24 hours of reperfusion may be too short to demonstrate an increased vascularization. In contrast, the loss of microvessels in the perinodal adipose tissue was less pronounced, providing further evidence that this flap component is less susceptible to IR-induced tissue injury.

Of interest, we detected an activation of eNOS in IR-exposed perinodal adipose tissue. This result supports the view of Gust et al., who recently suggested an active role of adipose tissue in driving the inflammatory response after IR injury [26]. They showed an overexpression of stress and inflammatory markers as well as inflammatory cell infiltration of mature adipose tissue following IR injury in a mouse model. Other authors found that severe ischemia is associated with loss of mature adipocytes, which are replaced with new adipocytes derived from resident adipose-derived progenitor cells [27]. These progenitor cells contribute to adipogenesis, angiogenesis and, presumably, also lymphangiogenesis. Despite moderate changes in cellularity and microvessel density observed in the present investigation, it may be assumed that the perinodal adipose tissue of VLN flaps exerts angiogenic and lymphangiogenic effects during the initial days to weeks after transplantation, contributing to the formation of a microvascular and lymphatic capillary network of VLN flaps. Remarkably, subsequent immunohistochemical analyses of the perinodal adipose tissue did not reveal a higher percentage of HO-1[+], PCNA[+] or Casp-3[+] cells. Furthermore, the number of infiltrating inflammatory cells was not markedly higher after 45 or 120 minutes of ischemia compared with the control group. This may be explained with technical differences of the experimental models. For instance, Gust et al. used an IR model with magnetic skin compression [26]. In contrast, the herein used IR model exposes the adipose tissue to indirect ischemia without mechanical manipulation. Hence, the less traumatic induction of ischemia might also result in fewer IR-associated tissue alterations.

Finally, immunohistochemical analyses revealed that lymph nodes of VLN flaps are prone to IR injury, as indicated by a higher percentage of HO-1[+] cells after 120 minutes ischemia. Moreover, lymph nodes undergoing 120 minutes ischemia also contained markedly higher numbers of proliferating PCNA[+] and apoptotic Casp-3[+] cells as well as CD68[+] macrophages and MPO[+] neutrophilic granulocytes when compared to lymph nodes of the IR-45 and control group. In contrast to our findings, the current opinion on critical ischemia time of VLN flaps is four hours as evaluated for inguinal VLN flaps in a rat model [28,29]. However, a recent analysis of inflammatory biomarkers revealed that significant IR injury can occur with as little as two hours of ischemia [30]. In line with this, the histological and immunohistochemical findings of the present study support the assumption that lymph nodes may be more susceptible to IR injury as previously assumed and already undergo significant histological changes after two hours of ischemia.

This study has important limitations. First, we only analyzed short-term cellular and molecular effects of IR injury on VLN flaps. The herein used animal model is not suitable to investigate the long-term effect of upregulated growth factors or cell damage on the integration and function of VLN flaps. This research question must be addressed in future experiments. Second, we only analyzed a small selection of growth factors involved in VLN flap function. Nevertheless, our findings contribute to the understanding of the initial phase of IR injury in lymph nodes and perinodal adipose tissue. Finally, in clinical practice, VLN flaps are performed with microsurgical anastomoses. In the present study, we used *in situ* clamping of the vascular pedicle to induce IR injury. However, in the experimental setting and particularly when working with rodent models, this technique may be more reliable compared with microsurgical VLN flap transfer because it eliminates the potential bias of microvascular complications.

In summary, the present study demonstrates that VLN flaps are remarkably prone to IR injury. In particular, prolonged intraoperative ischemia of 120 minutes is already associated with a significant reduction of cellularity and vascularization of lymph nodes. Moreover, we found an increased expression of the growth factors VEGF-D and VEGF-A as well as apoptotic cell death and immune cell infiltration of the nodal flap component. In contrast, the perinodal adipose tissue is less susceptible to IR injury but contributes to lymphangiogenic and angiogenic growth factor expression immediately after VLN flap surgery. From a translational point of view, further investigations are required to elucidate the long-term effect of IR injury on the functional capacity of VLN flaps.

## Material and methods

### Animals

The experiments were performed with 44 Lewis rats (Janvier Labs, Le Genest-Saint-Isle, France) exhibiting an age of $17 \pm 1$ weeks and a body weight of $397 \pm 8$ g. The animals were housed one per cage under a 12-hour day/night cycle and were fed ad libitum with water and standard pellet food (Altromin, Lage, Germany). All experiments were conducted in accordance with the European legislation on the protection of animals (Directive 2010/63/EU) and the National Institutes of Health guidelines on the care and use of laboratory animals (National Institutes of Health publication #85–23 Rev. 1985). They were approved by the local governmental animal protection committee (Landesamt für Verbraucherschutz, Saarbrücken, Germany; Permission number: 67/2015).

### Surgery

Flap dissection was performed by modifying a recently published model [31]. The rats were anesthetized by intraperitoneal injection of ketamine (80 mg/kg body weight; Ursotamin; Serumwerk Bernburg AG, Bernburg, Germany) and xylazine (6 mg/kg body weight; Rompun; Bayer, Leverkusen, Germany). After depilation, the animals were placed in supine position on a heated operation stage. After a skin incision in the axilla (Fig 1A), the pectoralis major muscle was retracted and the axillary neurovascular bundle was identified. Following identification of the lateral thoracic vessels (Fig 1B), the adjacent lymph node flap was circumferentially dissected with destruction of all afferent and efferent lymphatic vessels. Care was taken to preserve the brachial plexus. After complete dissection of the flap (Fig 1C), the lateral thoracic artery and vein (i.e. the flap pedicle) were clamped *in situ* using microvascular clamps (S&T AG Microsurgical Instruments, Neuhausen, Switzerland) according to the experimental protocol. After releasing the clamps and verification of the establishment of blood flow, the incision was closed with 5/0 monofilament (Prolene; Ethicon, Johnson & Johnson Medical GmbH, Norderstedt, Germany). Postoperative analgesia was provided by subcutaneous buprenorphine (0.05 mg/kg body weight; Buprenovet; Bayer Vital GmbH, Leverkusen, Germany) and tramadol hydrochloride (40 mg per 100 mL drinking water; Tramal; Grünenthal GmbH, Aachen, Germany). Wounds and animal behaviour were regularly checked until the rats were killed 24 hours after surgery.

### Ultrasound, photoacoustic imaging and stereomicroscopy

Rats with a dissected axillary VLN flap were anesthetized with 1.5% isoflurane and put on a heated stage. After covering the flap with ultrasound coupling gel (Aquasonic 100; Parker, Fairfield, NJ, USA), photoacoustic imaging was performed by means of a Vevo LAZR system (FUJIFILM Visualsonics Inc., Toronto, ON, Canada) and a real-time microvisualization

LZ550 linear-array transducer (FUJIFILM Visualsonics Inc.) with a center frequency of 40 MHz. Heart and breathing rate were constantly monitored and the body temperature was maintained at 36˚C (THM100; Indus Instruments, Houston, TX, USA). For three-dimensional imaging, the ultrasound probe was driven over the entire flap by a linear motor to acquire two-dimensional images at parallel and uniformly spaced, 150 μm-sized intervals. The two-dimensional image planes were then stitched together enabling rapid three-dimensional image reconstruction, displaying a dynamic cube view format, as described previously [32]. In addition, Oxy-Hemo-mode photoacoustic images were taken at two wavelengths (i.e. 750 nm and 850 nm) with a two-dimensional photoacoustic gain of 44 dB and a hemoglobin threshold of 20 dB. To measure the total hemoglobin signal [$HbT/mm^3$] within lymph nodes, all detected signals at the two wavelengths where divided by the volume of the analyzed lymph nodes. Furthermore, the oxygen saturation [$sO_2$] of individual lymph nodes was evaluated in %, as described previously [33,34]. All data values were analyzed using the Vevo LAB 1.7.2 software (FUJIFILM Visualsonics Inc.).

For stereomicroscopy, excised VLN flaps were placed under a stereomicroscope (Leica M651; Leica, Wetzlar, Germany) and the recorded images were transferred to a DVD system.

## Western blot

After shock freezing in liquid nitrogen and storage at—80˚C, the lymph node and adipose tissue samples were lysed by homogenization using a lysis buffer to extract the whole cell protein fraction with additional protease inhibitors (0.5 mM phenylmethylsulfonyl fluoride, 1:75 v/v Protease Inhibitor Cocktail, 1:100 v/v Phosphatase Inhibitor Cocktail 2; Sigma-Aldrich, Taufkirchen, Germany). Lysates were then collected and centrifuged at 4˚C and 13 000 xg for 30 minutes. Supernatants were saved as whole protein extracts and protein concentrations were analyzed photometrically using the Lowry method. Then, 30 μg protein per lane were separated on 10% sodium dodecyl sulfate polyacrylamide gels and transferred to a polyvinylidene difluoride membrane (Bio-Rad Laboratories, München, Germany). Subsequently, membranes were incubated with the following primary antibodies over night at 4˚C, followed by a supplemental incubation for 3 hours at room temperature: VEGF-C (1:300; Abcam, Cambridge, UK), VEGF-D (1:300; Abcam), VEGF-A (1:100; Santa Cruz Technology, Heidelberg, Germany), iNOS (1:300; Abcam), eNOS (1:300; BD Biosciences, Heidelberg, Germany) and p-eNOS (1:500; Cell Signaling Technology, Frankfurt, Germany). Corresponding horseradish peroxidase-conjugated secondary antibodies (1:5 000; GE Healthcare, Freiburg, Germany or 1:1 000; R&D Systems, Wiesbaden, Germany) were then attached at room temperature for 1.5 hours. Protein expression was visualized with enhanced chemiluminescence (ECL Western Blotting Analysis System, GE Healthcare) and analyzed with an ECL ChemoCam Imager (Chemostar and LabImage 1D software; Intas Science Imaging Instruments, Göttingen, Germany).

## Histology and immunohistochemistry

After excision and stereomicroscopy, flaps were fixed in 4% formalin, embedded in paraffin, and cut into 3-μm thick sections. Individual sections were stained with hematoxylin and eosin (HE) according to standard procedures. Using a BX60 microscope (Olympus, Hamburg, Germany) and the imaging software cellSens Dimension 1.11 (Olympus), the cellularity (number of nuclei, given in $mm^{-2}$) of individual lymph nodes was assessed at 800x magnification in 5 randomized regions of interest (ROIs) per section and of the perinodal adipose tissue at 200x magnification in 4 randomized ROIs per section.

For the immunohistochemical detection of lymphatic vessels, additional sections were stained with a polyclonal rabbit antibody against lymphatic vessel endothelial hyaluronan receptor-1 (LYVE-1) (1:600; Abcam). A goat anti-rabbit IgG Alexa555 antibody (1:200; Life Technologies, Eugene, OR, USA) served as secondary antibody. Cell nuclei were stained with Hoechst 33342 (2 µg/mL; Sigma-Aldrich) for image merging.

For the immunohistochemical detection of microvessels, sections were stained with a monoclonal rat antibody against the endothelial cell marker CD31 (1:100; Dianova, Hamburg, Germany). A goat anti-rat IgG Alexa555 antibody (1:200; Thermo Fisher Scientific GmbH, Dreieich, Germany) was used as secondary antibody. Cell nuclei were stained with Hoechst 33342 (2 µg/mL; Sigma-Aldrich) for image merging. The density of CD31$^+$ microvessels (given in mm$^{-2}$) of individual lymph nodes was assessed at 400x magnification within 5 randomized ROIs and of the perinodal adipose tissue at 200x magnification within 4 randomized ROIs per section.

Further sections were stained with a polyclonal rabbit antibody against HO-1 (1:100; Enzo Life Sciences GmbH, Lörrach, Germany). A peroxidase-labeled goat anti-rabbit IgG antibody (1:200; Dianova) served as secondary antibody. Additional sections were stained with a monoclonal mouse antibody PCNA (1:100; Agilent, Hamburg, Germany). A peroxidase-labeled goat anti-mouse IgG antibody (1:200; Dianova) served as secondary antibody. Finally, sections were stained with a polyclonal rabbit antibody against Casp-3 (1:100; New England BioLabs GmbH, Frankfurt, Germany) followed by a biotinylated goat anti-rabbit IgG antibody (ready-to-use; Abcam). The biotinylated antibodies were detected by peroxidase-labeled streptavidin (ready-to-use; Abcam). 3-Amino-9-ethylcarbazole (Abcam) was used as chromogen for PCNA-stained sections and 3,3-diaminobenzidine (Sigma-Aldrich) was used as chromogen for Casp-3 and HO-1-stained sections. The number of HO-1$^+$, PCNA$^+$ and Casp-3$^+$ cells (given in % of all cells) of individual lymph nodes was analyzed at 400x magnification in 6 randomized ROIs per section and of the perinodal adipose tissue at 200x magnification in 4 randomized ROIs per section.

Finally, sections were stained with a polyclonal rabbit antibody against CD68 (1:50; Abcam) and a polyclonal rabbit antibody against MPO (1:100, Abcam). This was followed by a biotinylated goat anti-rabbit IgG antibody (ready-to-use; Abcam), which was detected by peroxidase-labeled streptavidin (ready-to-use; Abcam). 3-Amino-9-ethylcarbazole (Abcam) was used as chromogen. Cell infiltration (given in mm$^{-2}$) of individual lymph nodes was quantified at 400x magnification in 5 randomized ROIs per section and of the perinodal adipose tissue at 200x magnification in 4 randomized ROIs per section. All quantitative analyses were performed with the software package ImageJ.

## Experimental protocol

For photoacoustic validation of the animal model, axillary VLN flaps of 8 rats were scanned before ischemia, at the beginning of ischemia, after 45 minutes (n = 4) or 120 minutes (n = 4) of ischemia as well as directly, 10 minutes and 24 hours after reperfusion. The animals were sacrificed after the last scan.

For histology and immunohistochemistry, VLN flaps were dissected in 24 rats. In the control group (n = 8), no ischemia was applied and the wounds were directly closed after complete flap dissection. In the IR-45 (n = 8) and IR-120 group (n = 8), the vascular pedicle was clamped for 45 and 120 minutes, respectively. The animals were sacrificed 24 hours after skin suture (control) or onset of reperfusion (IR-45/IR-120) and tissue specimens were processed for histological and immunohistochemical analyses.

For Western blotting, VLN flaps were dissected in 12 rats. Following the IR protocol for the control (n = 4), IR-45 (n = 4) and IR-120 (n = 4) group, the animals were sacrificed and the lymph nodes and the perinodal adipose tissue were excised and processed for protein expression analyses.

## Statistics

Data were analyzed for normal distribution and equal variance. Parametric data were compared using one-way analysis of variance (ANOVA) followed by Bonferroni's post hoc test. In case of non-parametric distribution, groups were analyzed with the Kruskal-Wallis test followed by Dunn's post hoc test. To test for time effects within groups, ANOVA for repeated measurements was applied followed by Tukey's post hoc test. Data are given as scatter dot plots with mean ± standard error of the mean (SEM). Statistical significance was accepted for $P < 0.05$. The statistical analysis was performed using Prism 7.0d (GraphPad Software, Inc.).

## Supporting information

**S1 Raw images.**
(PDF)

## Acknowledgments

We are grateful for the excellent technical assistance of Janine Becker, Caroline Bickelmann, Ruth Nickels and Julia Parakenings.

## Author Contributions

**Conceptualization:** Florian S. Frueh, Yves Harder, Michael D. Menger, Matthias W. Laschke.

**Data curation:** Florian S. Frueh, Bijan Jelvani, Claudia Scheuer, Christina Körbel, Emmanuel Ampofo.

**Formal analysis:** Florian S. Frueh, Bijan Jelvani, Claudia Scheuer, Christina Körbel, Bong-Sung Kim, Emmanuel Ampofo, Michael D. Menger, Matthias W. Laschke.

**Funding acquisition:** Florian S. Frueh.

**Investigation:** Florian S. Frueh, Bijan Jelvani, Claudia Scheuer, Christina Körbel.

**Methodology:** Florian S. Frueh, Emmanuel Ampofo, Michael D. Menger, Matthias W. Laschke.

**Project administration:** Florian S. Frueh.

**Visualization:** Bijan Jelvani, Claudia Scheuer, Christina Körbel.

**Writing – original draft:** Florian S. Frueh, Claudia Scheuer, Christina Körbel.

**Writing – review & editing:** Bong-Sung Kim, Pietro Giovanoli, Nicole Lindenblatt, Yves Harder, Emmanuel Ampofo, Michael D. Menger, Matthias W. Laschke.

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
