## [Decision Letter · Decision Letter 0]

27 Sep 2019

PONE-D-19-21979

Molecular and cellular effects of ischemia/reperfusion on vascularized lymph node flaps in rats

PLOS ONE

Dear Dr. Frueh,

Thank you for submitting your manuscript to PLOS ONE. After careful consideration, we feel that it has merit but does not fully meet PLOS ONE’s publication criteria as it currently stands. Therefore, we invite you to submit a revised version of the manuscript that addresses the points raised during the review process.

Please address the reviewer's points.

We would appreciate receiving your revised manuscript by Nov 11 2019 11:59PM. To enhance the reproducibility of your results, we recommend that if applicable you deposit your laboratory protocols in protocols.io, where a protocol can be assigned its own identifier (DOI) such that it can be cited independently in the future. For instructions see: http://journals.plos.org/plosone/s/submission-guidelines#loc-laboratory-protocols

We look forward to receiving your revised manuscript.

Kind regards,

Ferenc Gallyas, Jr., Ph.D., D.Sc.

Academic Editor

PLOS ONE

Journal Requirements:

Reviewers' comments:

Reviewer's Responses to Questions

**Comments to the Author**

1. Is the manuscript technically sound, and do the data support the conclusions?

Reviewer #1: Partly

2. Has the statistical analysis been performed appropriately and rigorously? 

Reviewer #1: Yes

3. Have the authors made all data underlying the findings in their manuscript fully available?

Reviewer #1: Yes

4. Is the manuscript presented in an intelligible fashion and written in standard English?

Reviewer #1: Yes

5. Review Comments to the Author

Reviewer #1: The authors evaluated the molecular and cellular effects of ischemia/reperfusion on vascularized lymph node flap in a rat model. After ischemia treatment of 45 min or 120 min and followed by reperfusion for 24 hours, the oxygen content, level of oxidative stress and growth factors were examined. The authors separate lymph nodes and perinodal adipose tissue to observe the ischemia/reperfusion effects. The number of proliferating and apoptotic cells were increased after ischemia/reperfusion treatment. Some data still required to be added to make the manuscript more complete.

Specific comments:

a. In Fig 3, the authors observed the expression of LYVE. Did any changes in the expression of LYVE under different ischemia time or in the location of LYVE expression? Any scientific relevant of using LYVE in this staining, since the authors only observed the structure of the lymph node under ischemia treatment.

b. The results on Fig. 4 presented the increase level of VEGF-A. Did the authors stain for CD31 or other angiogenesis markers to evaluate whether the blood vessel density were also increased in lymph nodes or adipose tissue?

c. The increase number of proliferating cells were identified in Fig. 5 by using PCNA+ marker. However, the other results indicated the cellularity were decreased after 45 min/120 min ischemia and 24 hours reperfusion treatment (in Fig. 3). What kind of cells were the proliferating cells? And what kind of cell were the caspas 3 labeled?

d. How the authors choose the 45 and 120 min as the ischemia time and 24 hours as the reperfusion time?

e. In the Discussion, the authors emphasized on the different effects after ischemia treatment on lymph nodes and adipose tissue. However, the ischemia effect on lymph nodes were presented in the Fig. 4. There were no cell cellularity, HO-1+, PCNA+ or immune cell examination on adipose tissue.

6. PLOS authors have the option to publish the peer review history of their article (what does this mean?). If published, this will include your full peer review and any attached files.

Reviewer #1: No

---

## [Author Response · Author response to Decision Letter 0]

26 Jan 2020

Review of the manuscript PONE-D-19-21979 

Reply to the comments of reviewer 1

Reviewer #1: The authors evaluated the molecular and cellular effects of ischemia/reperfusion on vascularized lymph node flap in a rat model. After ischemia treatment of 45 min or 120 min and followed by reperfusion for 24 hours, the oxygen content, level of oxidative stress and growth factors were examined. The authors separate lymph nodes and perinodal adipose tissue to observe the ischemia/reperfusion effects. The number of proliferating and apoptotic cells were increased after ischemia/reperfusion treatment. Some data still required to be added to make the manuscript more complete.

We thank the reviewer for the fair and constructive comments. Please find our point-by-point reply in the following.

Specific comments: 

Reviewer comment 1

In Fig 3, the authors observed the expression of LYVE. Did any changes in the expression of LYVE under different ischemia time or in the location of LYVE expression? Any scientific relevant of using LYVE in this staining, since the authors only observed the structure of the lymph node under ischemia treatment.

We thank the reviewer for this important comment. The reviewer is correct that LYVE-1 staining in this part of our investigation was only used for descriptive purposes and not for a quantitative analysis of lymphatic vessel density. In this context it should be noted that the expression of LYVE-1 in the lymphatic sinuses of lymph nodes is not limited to lymphatic endothelial cells but is also observed in other cell types, such as reticular cells (Bai et al., 2011). Consequently, a quantification of the lymphatic vascular network (ie, the sinuses) within individual lymph nodes is not appropriate using LYVE-1 as marker. 

Reference

Bai Y, Wu B, Terada N, Ohno N, Saitoh S, Saitoh Y, Ohno S. Histological study and LYVE-1 immunolocalization of mouse mesenteric lymph nodes with "in vivo cryotechnique". Acta Histochem Cytochem. 2011;44:81-90.

Reviewer comment 2

The results on Fig. 4 presented the increase level of VEGF-A. Did the authors stain for CD31 or other angiogenesis markers to evaluate whether the blood vessel density were also increased in lymph nodes or adipose tissue?

According to the comment of the reviewer, we have performed additional immunohistochemical stainings of VLN flaps using the endothelial cell marker CD31 to quantify the microvessel density (mm-2) within individual lymph nodes and the perinodal adipose tissue. Interestingly, lymph nodes undergoing 45 and 120 minutes of ischemia exhibited a significantly reduced microvessel density when compared to non-ischemic controls. In contrast, the microvascular network of the perinodal adipose tissue was less sensitive to ischemia/reperfusion injury. This novel information is now included in the revised version of our manuscript (see Figure 4; Results, page 6/7, lines 171-188 as well as Discussion page 12/13, lines 341-347; marked in yellow)

Reviewer comment 3

The increase number of proliferating cells were identified in Fig. 5 by using PCNA+ marker. However, the other results indicated the cellularity were decreased after 45 min/120 min ischemia and 24 hours reperfusion treatment (in Fig. 3). What kind of cells were the proliferating cells? And what kind of cell were the caspas 3 labeled?

The reviewer is correct that despite an overall reduction of cellularity, we observed an increasing number of proliferating cells in the lymph nodes of ischemic VLN flaps. However, this is a common finding after ischemia/reperfusion injury. Due to technical issues, we were not able to further characterize PCNA+ and caspase-3+ cells with co-stainings using lymphocyte-specific antibodies. Instead, we have performed a morphological analysis of individual PCNA+ and caspase-3+ cells within ischemic lymph nodes and found that the majority of them were lymphocytes. This is in line with the findings of a recently published study by Perrault et al. (2020), who reported a particularly high sensitivity of the immune cell component of lymph nodes to ischemia reperfusion injury. This novel information is now included in the revised version of our manuscript (see Results, page 9, lines 234-236; marked in yellow).

Reference

Perrault DP, Lee GK, Bouz A, Sung C, Yu R, Pourmoussa AJ, Park SY, Kim GH, Jiao W, Patel KM, Hong YK, Wong AK. Ischemia and reperfusion injury in superficial inferior epigastric artery-based vascularized lymph node flaps. PLoS One. 2020;15:e0227599.

Reviewer comment 4

How the authors choose the 45 and 120 min as the ischemia time and 24 hours as the reperfusion time?

We chose the ischemia time based on realistic clinical scenarios. In fact, 45 minutes represent an uncomplicated microvascular VLN flap transplantation and 120 minutes represent a difficult or suboptimal microvascular procedure. The reperfusion time was chosen based on our previous experience with ischemia/reperfusion experiments and it has recently been shown that shorter reperfusion times may reduce the biological reaction to ischemia of VLN flaps (Perrault et al., 2020). 

Reference

Perrault DP, Lee GK, Bouz A, Sung C, Yu R, Pourmoussa AJ, Park SY, Kim GH, Jiao W, Patel KM, Hong YK, Wong AK. Ischemia and reperfusion injury in superficial inferior epigastric artery-based vascularized lymph node flaps. PLoS One. 2020;15:e0227599.

Reviewer comment 5

In the Discussion, the authors emphasized on the different effects after ischemia treatment on lymph nodes and adipose tissue. However, the ischemia effect on lymph nodes were presented in the Fig. 4. There were no cell cellularity, HO-1+, PCNA+ or immune cell examination on adipose tissue.

According to the comment of the reviewer, we have performed an additional immunohistochemical analysis of the perinodal adipose tissue, evaluating the fraction of HO-1+, PCNA+ and caspase-3+ cells as well as the number of infiltrating CD68+ macrophages and MPO+ neutrophilic granulocytes. Of interest, we found that the cells within the perinodal adipose tissue were not markedly affected by 45 or 120 minutes of ischemia. This indicates again a higher resistance of adipose tissue to ischemia/reperfusion injury when compared to the lymph node component of VLN flaps. This novel information is now included in the revised version of our manuscript (Figure 7; see Results, page 9, lines 236-239 and lines 247-250 as well as page 10, lines 267-281; see Discussion, page 13/14, lines 369-374; marked in yellow).

Two new co-authors from the Division of Plastic Surgery and Hand Surgery, University Hospital Zurich and from the Institute for Clinical & Experimental Surgery, Saarland University, markedly contributed to our novel immunohistochemical analyses. Accordingly, both co-workers have been included in the author list of the revised manuscript (see page 1, lines 6 and 7; see page 23, lines 632 and 636; marked in yellow).

---

## [Decision Letter · Decision Letter 1]

7 May 2020

PONE-D-19-21979R1

Molecular and cellular effects of ischemia/reperfusion on vascularized lymph node flaps in rats

PLOS ONE

Dear Dr. Frueh,

Thank you for submitting your manuscript to PLOS ONE. After careful consideration, we feel that it has merit but does not fully meet PLOS ONE’s publication criteria as it currently stands. Therefore, we invite you to submit a revised version of the manuscript that addresses the points raised during the review process.

We would appreciate receiving your revised manuscript by Jun 20 2020 11:59PM. To enhance the reproducibility of your results, we recommend that if applicable you deposit your laboratory protocols in protocols.io, where a protocol can be assigned its own identifier (DOI) such that it can be cited independently in the future. For instructions see: http://journals.plos.org/plosone/s/submission-guidelines#loc-laboratory-protocols

We look forward to receiving your revised manuscript.

Kind regards,

Jianhong Zhou

Associate Editor

PLOS ONE

Reviewers' comments:

Reviewer's Responses to Questions

**Comments to the Author**

1. If the authors have adequately addressed your comments raised in a previous round of review and you feel that this manuscript is now acceptable for publication, you may indicate that here to bypass the “Comments to the Author” section, enter your conflict of interest statement in the “Confidential to Editor” section, and submit your "Accept" recommendation.

Reviewer #1: All comments have been addressed

Reviewer #2: (No Response)

2. Is the manuscript technically sound, and do the data support the conclusions?

Reviewer #1: Yes

Reviewer #2: Yes

3. Has the statistical analysis been performed appropriately and rigorously? 

Reviewer #1: Yes

Reviewer #2: Yes

4. Have the authors made all data underlying the findings in their manuscript fully available?

Reviewer #1: Yes

Reviewer #2: Yes

5. Is the manuscript presented in an intelligible fashion and written in standard English?

Reviewer #1: Yes

Reviewer #2: Yes

6. Review Comments to the Author

Reviewer #1: The authors evaluated the molecular and cellular effects of ischemia on vascularized lymph node flap. After ischemia of 45 min or 120 min and followed by reperfusion for 24 hours., the oxygen content, the level of oxidative stress and growth factors were examined. The authors separate lymph nodes and perinodal adipose tissue to observe the ischemia effect. The number of proliferating and apoptotic cells were increased after ischemia/reperfusion treatment. The authors made much revisions regarding the reviewers’ comments. Some questions remain as follows:

1. In the result section (page 5), “r and t = 900 μm“, and in the figure legend of Fig. 2, “R and T= 900 μm” are not compatible.

2. On page 6, “ In the IR-120 group, however, we observed a partial loss of the lymph node architecture (Figs. 3F and 3G)”, which particular part was lost?

3. In Fig. 4D, the CD 31 staining seems more abundant by compared to the other two groups, which was inconsistent to the results of Fig. 4 E.

4. According to the discussion, authors referenced Gust’s paper indicated “an active role of adipose tissue in driving the inflammatory response after IR injury [26]. The overexpression of stress and inflammatory markers as well as inflammatory cell infiltration of mature adipose tissue following IR injury. But other authors found that severe ischemia is associated with loss of mature adipocytes, which are replaced with new adipocytes derived from resident adipose-derived progenitor cells [27]. “

Since the results showing a higher level of eNOS presented in the perinodal adipose tiåssue in Fig. 6, there was no significance cell proliferation, apoptosis, no oxidative stress in Fig. 7, which were inconsistent with the results from the reference that the authors provided. How to explain these findings?

5. The results indicated an increasing level of VEGF-A after ischemia condition (Fig. 5). The results were not consistent with that in Fig. 4, in which the micro-vessel density was decreased after ischemia condition.

6. The staining of CD31 in Fig. 4D was not compatible with Figs. 4E and 4F.

Reviewer #2: This manuscript describes the results of a preclinical study analyzing the immediate molecular and cellular effects after ischemia-reperfusion injury on vascularized lymph node transfers. It analyzes various lymphangiogenic growth factors, as well as it evaluates the morphology, proliferation, apoptosis, and infiltration of immune cells in the lymph nodes and perinodal adipose tissue. For this study the authors have used a rat model of vascularized lymph node flap pedicled on the lateral thoracic vessels.

Overall, this is a good research set-up which is well described as well as conducted. However, there are some major and minor revisions to be done.

1. Indicate in the title, in the abstract and when the objective of the study is stated at the end of the introduction, that the main goal is to investigate the “immediate” or “initial” changes at the cellular and molecular level after performing lymph node transfers, since the study ends at 24h postop. Findings may be interpreted as biomarkers to identify the initial phase of I/R injury in lymph nodes and perinodal tissue.

2. The abstract does not indicate which animal model or which lymph nodes are studied, nor the number of animals, nor their distribution in the study groups. Please modify it accordingly.

3. Please modify the image Figure 1.B as it is not clearly seen.

4. In the discussion please focus more on your actual results and their interpretation. This section partially sounds like an introduction. You may move parts to the introduction if needed.

5. In the final paragraph of the discussion (conclusion) try to be more specific and summarize the significant results you have obtained in this study. This will help other researchers and readers to get the most relevant results of this study.

6. It is important that readers keep in mind the limitations of this study. Include a summary of the main ones and explain them in the discussion. Examples:

- There are numerous growth factors that were not included in the analysis, in this sense, we should not exclude the importance of other biomarkers.

- Vascularized lymph node transfers are performed using microsurgical anastomoses, however in this study, VLNTs have been simulated by clamping the vessels.

- The current literature indicates that the critical ischemia time in lymph node transfers is 4h, however in this study a shorter ischemic insult has been studied.

7. In the Material and Methods section, it is indicated that an injectable anesthetic protocol (Ketamine and xylazine) is used during surgery, but during the ultrasound, photoacoustic imaging and stereomicrsocopy evaluation, inhalation anesthesia with isoflurane is used. Why have you used different protocols?

8. In order to strictly adhere to the three R's principle, why have you not used the 8 rats that were used for photoacoustic validation for histology, immunohistochemistry or Western blot studies?

7. PLOS authors have the option to publish the peer review history of their article (what does this mean?). If published, this will include your full peer review and any attached files.

Reviewer #1: No

Reviewer #2: Yes: Alberto Ballestín

---

## [Author Response · Author response to Decision Letter 1]

1 Jun 2020

Review of the manuscript PONE-D-19-21979R1 by Frueh et al.

We have appreciated the fair and constructive comments of the reviewers. Please find our point-by-point reply in the following.

Reply to the comments of reviewer 1

Reviewer comment 1:

In the result section (page 5), “r and t = 900 μm“, and in the figure legend of Fig. 2, “R and T= 900 μm” are not compatible.

We have corrected the corresponding section according to the reviewer’s comment (see page 6, line 151, marked in yellow).

Reviewer comment 2:

On page 6, “ In the IR-120 group, however, we observed a partial loss of the lymph node architecture (Figs. 3F and 3G)”, which particular part was lost?

Our statement was only based on qualitative assessments of lymph node morphology. However, we did not further analyze whether specific parts of the lymph node architecture were lost. Accordingly, we have removed this vague statement from our revised manuscript and now solely focus on the quantitative analysis of lymph node cellularity (see page 6, lines 156-159 and page 12, line 306-312; marked in yellow).

Reviewer comment 3:

In Fig. 4D, the CD 31 staining seems more abundant by compared to the other two groups, which was inconsistent to the results of Fig. 4 E.

The reviewer is correct that the CD31 staining in Fig. 4D looks more abundant. However, the red area in the center of the lymph node is due to local hemorrhages (ie, red stained clusters of erythrocytes). Importantly, microvessels characterized by a CD31+ endothelium are marked with arrowheads for the reader. In addition, the hemorrhages are now marked with separate arrows (see revised Fig. 4D and page 7, line 190, marked in yellow).

Reviewer comment 4:

According to the discussion, authors referenced Gust’s paper indicated “an active role of adipose tissue in driving the inflammatory response after IR injury [26]. The overexpression of stress and inflammatory markers as well as inflammatory cell infiltration of mature adipose tissue following IR injury. But other authors found that severe ischemia is associated with loss of mature adipocytes, which are replaced with new adipocytes derived from resident adipose-derived progenitor cells [27]. “

Since the results showing a higher level of eNOS presented in the perinodal adipose tissue in Fig. 6, there was no significance cell proliferation, apoptosis, no oxidative stress in Fig. 7, which were inconsistent with the results from the reference that the authors provided. How to explain these findings?

We appreciate this important comment. Indeed, it is noteworthy that in our investigation, the adipose tissue was characterized by a less dramatic reaction to IR injury. However, a more thorough look at the referenced animal model by Gust et al. reveals important technical differences when compared to our experimental model. Importantly, Gust et al. used an IR model with magnetic skin compression. In contrast, our IR model exposes the adipose tissue to indirect ischemia without mechanical manipulation. Hence, the less extensive tissue trauma might also result in fewer IR-associated tissue alterations. We now provide this explanation in the discussion section of our revised manuscript (see page 13-14, lines 365-373, marked in yellow).

Reviewer comment 5:

The results indicated an increasing level of VEGF-A after ischemia condition (Fig. 5). The results were not consistent with that in Fig. 4, in which the micro-vessel density was decreased after ischemia condition.

From a biological point of view, this finding is not surprising because IR-related tissue damage leads to the loss of microvascular structures and local hypoxia. Consequently, angiogenic growth factors such as VEGF-A are upregulated, promoting angiogenic neovessel growth. However, angiogenesis is a time-consuming process with a slow growth rate of new microvessels of 5 µm per hour. Hence, 24 hours of reperfusion may be too short to demonstrate a subsequent increased vascularization. This information is now provided in the revised discussion of our manuscript (see page 13, lines 345-352 and page 23, lines 627-628; marked in yellow).

New reference

Utzinger U, Baggett B, Weiss JA, Hoying JB, Edgar LT. Large-scale time series microscopy of neovessel growth during angiogenesis. Angiogenesis. 2015;18:219-232.

Reviewer comment 6:

The staining of CD31 in Fig. 4D was not compatible with Figs. 4E and 4F.

See reply to comment 4.

Reply to the comments of reviewer 2

Reviewer comment 1:

Indicate in the title, in the abstract and when the objective of the study is stated at the end of the introduction, that the main goal is to investigate the “immediate” or “initial” changes at the cellular and molecular level after performing lymph node transfers, since the study ends at 24h postop. Findings may be interpreted as biomarkers to identify the initial phase of I/R injury in lymph nodes and perinodal tissue.

We thank the reviewer for this important comment. According to his suggestion, we now state in the title, abstract and introduction that our study investigates the short-term molecular and cellular effects of I/R on VLN (see page 1, line 3 and page 2, line 41 and page 4, line 99 and page 11, line 296, marked in yellow).

Reviewer comment 2:

The abstract does not indicate which animal model or which lymph nodes are studied, nor the number of animals, nor their distribution in the study groups. Please modify it accordingly.

According to this comment of the reviewer, we now provide the total number of animals used in the study as well as more details on the animal model in the abstract (see page 2, lines 42-44, marked in yellow). 

However, to enhance the readability of the manuscript, we prefer not to mention the distribution of the animals in the study groups in the abstract. In fact, this essential information is already provided in two sections of our manuscript (ie, experimental protocol and figure legends). 

Reviewer comment 3:

Please modify the image Figure 1.B as it is not clearly seen.

We have modified the figure including a more detailed explanation in the corresponding figure legend (see revised Figs. 1B and 1C and corresponding figure legend, page 4, lines 119-124, marked in yellow).

Reviewer comment 4:

In the discussion please focus more on your actual results and their interpretation. This section partially sounds like an introduction. You may move parts to the introduction if needed.

According to this comment of the reviewer, we have moved parts of the discussion to the introduction and focused more on our results (see page 3, lines 87-94 and pages 11-14, lines 306-385; marked in yellow).

Reviewer comment 5:

In the final paragraph of the discussion (conclusion) try to be more specific and summarize the significant results you have obtained in this study. This will help other researchers and readers to get the most relevant results of this study.

According to this comment, we have modified the last paragraph of the discussion in our revised manuscript (see page 14-15, lines 398-406, marked in yellow).

Reviewer comment 6:

It is important that readers keep in mind the limitations of this study. Include a summary of the main ones and explain them in the discussion. Examples:

- There are numerous growth factors that were not included in the analysis, in this sense, we should not exclude the importance of other biomarkers.

- Vascularized lymph node transfers are performed using microsurgical anastomoses, however in this study, VLNTs have been simulated by clamping the vessels.

- The current literature indicates that the critical ischemia time in lymph node transfers is 4h, however in this study a shorter ischemic insult has been studied.

We thank the reviewer for this comment. According to his suggestion, we have included a novel paragraph on the limitations of our study in the discussion section of our revised manuscript (see page 14, lines 386-397, marked in yellow).

Reviewer comment 7:

In the Material and Methods section, it is indicated that an injectable anesthetic protocol (Ketamine and xylazine) is used during surgery, but during the ultrasound, photoacoustic imaging and stereomicroscopy evaluation, inhalation anesthesia with isoflurane is used. Why have you used different protocols?

The reviewer is correct that flap dissection was performed under intraperitoneal anesthesia using ketamine and xylazine. After transfer and fixation of the animals in the ultrasound/photoacoustic imaging system, maintenance of anesthesia was performed using isoflurane inhalation under continuous monitoring of heart and breathing rate as well as body temperature. On the one hand, inhalation anesthesia is suitable to precisely control the depth of anesthesia. On the other, the animal is fixed in a complex imaging system, which renders intraperitoneal injection of anesthetics cumbersome and a source of bias due to animal manipulation, particularly during photoacoustic imaging.

Reviewer comment 8

In order to strictly adhere to the three R's principle, why have you not used the 8 rats that were used for photoacoustic validation for histology, immunohistochemistry or Western blot studies?

We thank the reviewer for this important comment. However, it can be answered by maximal standardization of the perioperative protocol. Including these rats in the groups for histology, immunohistochemistry or Western blot analyses would not only have resulted in a different protocol (ie, transfer and delay for photoacoustic imaging), it would also be biased by different anesthesia medications as correctly stated by the reviewer in comment 7. Finally, we performed photoacoustic imaging for the IR-45 and IR-120 groups but not for the control group. Hence, inclusion of these animals for subsequent histological, immunohistochemical or Western blot analyses would have also resulted in uneven numbers of the experimental groups.

---

## [Decision Letter · Decision Letter 2]

14 Aug 2020

PONE-D-19-21979R2

Short-term molecular and cellular effects of ischemia/reperfusion on vascularized lymph node flaps in rats

PLOS ONE

Dear Dr. Frueh,

Thank you for submitting your manuscript to PLOS ONE. After careful consideration, we feel that it has merit but does not fully meet PLOS ONE’s publication criteria as it currently stands. Therefore, we invite you to submit a revised version of the manuscript that addresses the points raised during the review process.

We look forward to receiving your revised manuscript.

Kind regards,

Zhejun Cai, M.D.

Academic Editor

PLOS ONE

Reviewers' comments:

Reviewer's Responses to Questions

6. Review Comments to the Author

Reviewer #1: This is the revision manuscript. Authors tried to answers most of reviewer’s comments.

Many inconsistent statements had been corrected. The scientific concepts are clearer in this revision manuscript. There are few comments as follows.

a. In Fig 3, any difference on LYVE expression, since it’s the marker for lymphatic network.

b. In Fig.4, it will be more convicting when the images of CD31staining in perinodal adipose are provided.

Reviewer #2: Authors have adequately addressed the comments raised in the previous round of review. The authors have refined the article by specifying that the objective was to study "immediate" or "initial" changes at the cellular and molecular level after performing lymph node transfers. In addition, both the introduction and the discussion have been improved, as well as the main limitations of the preclinical study have been included.

It is a scientific paper with a good research set-up which is well explained as well as conducted. It describes the results of a preclinical study analyzing the immediate molecular and cellular effects after ischemia-reperfusion injury on vascularized lymph node transfers. For this study the authors have used a rat model of vascularized lymph node flap pedicled on the lateral thoracic vessels. The article analyzes various lymphangiogenic growth factors, as well as it evaluates the morphology, proliferation, apoptosis, and infiltration of immune cells in the lymph nodes and perinodal adipose tissue in this murine model. The study demonstrates that vascularized lymph node flaps are prone to ischemia reperfusion injury, being associated with a significant reduction of cellularity and vascularization of lymph nodes, as well as apoptotic cell death and immune cell infiltration.

---

## [Author Response · Author response to Decision Letter 2]

20 Aug 2020

We have appreciated the fair and constructive comments of the reviewer. Please find our point-by-point reply in the following.

Reply to the comments of reviewer 1

Reviewer comment 1

In Fig 3, any difference on LYVE expression, since it’s the marker for lymphatic network.

We thank the reviewer for this comment. However, we have already discussed this point in the first review round and hereby provide our original reply:

“The reviewer is correct that LYVE-1 staining in this part of our investigation was only used for descriptive purposes and not for a quantitative analysis of lymphatic vessel density. In this context it should be noted that the expression of LYVE-1 in the lymphatic sinuses of lymph nodes is not limited to lymphatic endothelial cells but is also observed in other cell types, such as reticular cells (Bai et al., 2011). Consequently, a quantification of the lymphatic vascular network (ie, the sinuses) within individual lymph nodes is not appropriate using LYVE-1 as marker.”

Reference

Bai Y, Wu B, Terada N, Ohno N, Saitoh S, Saitoh Y, Ohno S. Histological study and LYVE-1 immunolocalization of mouse mesenteric lymph nodes with "in vivo cryotechnique". Acta Histochem Cytochem. 2011;44:81-90.

Reviewer comment 2

In Fig.4, it will be more convicting when the images of CD31staining in perinodal adipose are provided.

According to this comment of the reviewer, we now provide immunofluorescent images illustrating the CD31+ microvascular network in the perinodal adipose tissue in Fig. 4 (see revised Fig. 4 as well as the corresponding figure legend on page 7, lines 187-190 in the revised version of our manuscript, marked in yellow).

---

## [Decision Letter · Decision Letter 3]

9 Sep 2020

Short-term molecular and cellular effects of ischemia/reperfusion on vascularized lymph node flaps in rats

PONE-D-19-21979R3

Dear Dr. Frueh,

We’re pleased to inform you that your manuscript has been judged scientifically suitable for publication and will be formally accepted for publication once it meets all outstanding technical requirements.

Kind regards,

Zhejun Cai, M.D.

Academic Editor

PLOS ONE

Additional Editor Comments (optional):

Reviewers' comments:

Reviewer's Responses to Questions

**Comments to the Author**

1. If the authors have adequately addressed your comments raised in a previous round of review and you feel that this manuscript is now acceptable for publication, you may indicate that here to bypass the “Comments to the Author” section, enter your conflict of interest statement in the “Confidential to Editor” section, and submit your "Accept" recommendation.

Reviewer #1: All comments have been addressed

---

## [Editor Report · Acceptance letter]

25 Sep 2020

PONE-D-19-21979R3 

Short-term molecular and cellular effects of ischemia/reperfusion on vascularized lymph node flaps in rats 

Dear Dr. Frueh:

I'm pleased to inform you that your manuscript has been deemed suitable for publication in PLOS ONE. Congratulations! Your manuscript is now with our production department. 

Kind regards, 

on behalf of

Dr. Zhejun Cai 

Academic Editor

PLOS ONE